# The Effects of Ionising and Non-Ionising Electromagnetic Radiation on Extracellular Matrix Proteins

**DOI:** 10.3390/cells10113041

**Published:** 2021-11-05

**Authors:** Ren Jie Tuieng, Sarah H. Cartmell, Cliona C. Kirwan, Michael J. Sherratt

**Affiliations:** 1Division of Cell Matrix Biology & Regenerative Medicine, School of Biological Sciences, Faculty of Biology, Medicine and Health, University of Manchester, Manchester M13 9PT, UK; renjie.tuieng@postgrad.manchester.ac.uk; 2Department of Materials, School of Natural Sciences, Faculty of Science and Engineering and The Henry Royce Institute, Royce Hub Building, University of Manchester, Manchester M13 9PL, UK; sarah.cartmell@manchester.ac.uk; 3Division of Cancer Sciences and Manchester Breast Centre, Oglesby Cancer Research Building, Manchester Cancer Research Centre, School of Medical Sciences, Faculty of Biology, Medicine and Health, University of Manchester, Manchester M20 4BX, UK; cliona.kirwan@manchester.ac.uk; 4Division of Cell Matrix Biology & Regenerative Medicine and Manchester Breast Centre, Faculty of Biology, Medicine and Health, University of Manchester, Manchester M13 9PT, UK

**Keywords:** ionising radiation, X-rays, ultraviolet (UV) radiation, extracellular matrix (ECM), skin, breast, radiotherapy

## Abstract

Exposure to sub-lethal doses of ionising and non-ionising electromagnetic radiation can impact human health and well-being as a consequence of, for example, the side effects of radiotherapy (therapeutic X-ray exposure) and accelerated skin ageing (chronic exposure to ultraviolet radiation: UVR). Whilst attention has focused primarily on the interaction of electromagnetic radiation with cells and cellular components, radiation-induced damage to long-lived extracellular matrix (ECM) proteins has the potential to profoundly affect tissue structure, composition and function. This review focuses on the current understanding of the biological effects of ionising and non-ionising radiation on the ECM of breast stroma and skin dermis, respectively. Although there is some experimental evidence for radiation-induced damage to ECM proteins, compared with the well-characterised impact of radiation exposure on cell biology, the structural, functional, and ultimately clinical consequences of ECM irradiation remain poorly defined.

## 1. Introduction

Radiation exists in two main forms: Electromagnetic (EM) radiation in the form of alternating electric and magnetic waves that propagate energy, and particle radiation consisting of accelerated particles such as electrons and protons. EM radiation can be broadly categorised as non-ionising and ionising. Both types may be encountered clinically or environmentally, with exposure having potentially positive or negative effects on tissues and organisms (Table 1). In the case of non-ionising radiation, exposure of skin to ultraviolet radiation (UVR), for example, may be beneficial, as a consequence of vitamin D production [1], or detrimental, due photoageing [2] and/or photocarcinogenesis [3]. UVR is considered non-ionising as it is, in general, not sufficiently energetic to remove electrons from biomolecules. In contrast, energetic, ionising electromagnetic radiation (X-rays and gamma rays) can remove electrons. The undoubted importance of controlled exposure to ionising EM radiation in medical diagnostic imaging [4] and radiotherapy [5,6] must be balanced against side effects such as secondary cancers or tissue fibrosis [7,8]. Other forms of radiation, which rely on charged particles (e.g., α, β, protons), can also interact with biological systems and are clinically important (such as in proton therapy and in cosmic radiation exposure for space exploration), but being non-electromagnetic, they lie outside the scope of this review. The reader is referred to an excellent review by Helm et al. [9].

Most investigations into the detrimental side effects of radiation on biological tissues have largely focused on cellular damage, and in particular, the sensitivity of DNA [27,28]. Whilst acute high radiation exposure may kill cells, it has become increasingly clear that lower doses may have sub-lethal effects that are complex, difficult to eliminate and delayed (persisting over long periods of time) [2,7,29,30]. Crucially, to understand the consequences of radiation exposure and hence to potentially prevent or reverse the damage, it is necessary to characterise the interactions of radiation with not only cells but also with their complex and dynamic extracellular environment. This review considers the consequences and causative mechanisms that drive electromagnetic radiation damage in biological tissues and in the extracellular matrix (ECM) in particular. Two clinical models of interest are discussed: skin exposed to UVR in sunlight and breast tissues exposed to diagnostic and therapeutic X-rays.

### Electromagnetic Radiation

UVR and X-rays/Gamma rays, both being part of the EM radiation spectrum (Figure 1), differ only in wavelength, frequency and energy. When a molecule absorbs EM radiation, it undergoes one of three possible transitions: electronic, vibrational, or rotational [31]. In general, electronic transitions require the largest amount of energy, followed by vibrational then rotational [32].

Ionising radiation is often more energetic than non-ionising radiation and, as a result, is more likely to induce electronic transitions of atoms and molecules. In electronic excitation, an electron absorbing the radiation transits into a higher electronic state, becoming less bounded to the nucleus and therefore more reactive [33]. If the radiation has sufficient energy, the electron can escape the coulomb attraction of the nucleus, and the molecule is ionised. In contrast, molecules undergoing rotational or vibrational transitions (generally caused by non-ionising UVR exposure) experience minimal changes in the stability of the electron-nucleus attraction, resulting in negligible chemical effects. Therefore, exposure to ionising and non-ionising radiation results in significantly distinct biological molecular effects.

## 2. Non-Ionising Radiation (UVR)

UVR is conventionally designated as three categories of increasing energy, UVA (315 –400 nm), UVB (280–315 nm), and UVC (10–280 nm) [34]. UVA and UVB are of particular biological interest as they comprise the UVR in sunlight at the Earth’s surface (UV-A: 95%, UV-B: 5%) [35]. In contrast, UVC is absorbed efficiently in the atmosphere by ozone and oxygen and thus plays no role in environmental UVR damage [36,37].

### 2.1. Absorption of Non-Ionising Radiation (UVR)

Molecules or regions of molecules that absorb UVR are referred to as UV chromophores. Biological systems are rich in UV chromophores, including DNA and some amino acid residues [38]. In DNA, the nucleotides thymine and cytosine absorb UVB to become electronically excited [39,40]. In proteins, the amino acid residues tyrosine (Tyr), tryptophan (Trp) and cystine (double-bonded cysteine) absorb UVR from sunlight [41,42], with an absorbance peak at 280 nm for Tyr and Trp and lower for cystine [43,44]. For Tyr and Trp, their benzene ring structure facilitates an electronic transition from the ground state to the singlet excitation state that requires photons in the UVB region (180–270 nm) [45,46]. The excited chromophores can then transfer their energy or donate an electron to O_2_, forming several reactive oxygen species (ROS) [16,47,48]. The excess energy can cleave intermolecular bonds, such as disulphide bonds, or facilitate the formation of pyrimidine dimers in DNA [49,50].

UVR damage in biological organisms is largely mediated indirectly via the photodynamic production of unstable ROS [51]. UVR exposure generates ROS via the reaction between the excited UV chromophores and molecular oxygen (O_2_) [2] (Figure 2). In brief, the excited UV chromophore reacts with O_2_ to produce, through electron transfer, either a superoxide anion radical (O_2_^−^) or singlet oxygen (^1^O_2_) through energy transfer. Superoxide dismutases, which are present in the cell [52] and the ECM [53], convert O_2_^−^ into hydrogen peroxide (H_2_O_2_). In the presence of Fe(II), H_2_O_2_ undergoes the Fenton reaction to generate hydroxyl radicals (HO·) [2,54]. The cellular effects of both ^1^O_2_ and HO· are well studied [47,49,54]. Intracellular ROS have been shown to react with and cause damage to both proteins and DNA [55,56].

### 2.2. Biological Consequences of UVR Exposure

Intracellularly, UV-B photons can be absorbed directly by the DNA nucleotides thymine and cytosine to form cyclobutane pyrimidine dimers (CPDs) [50] and 6-4 photoproducts (6-4PP) [39]. These photoproducts can further absorb UV-A to form Dewar valence isomers [58]. CPDs, 6-4PP, and Dewar valence isomers are known as photolesions which disrupt the base pairing of DNA, preventing DNA transcription and replication [16]. Photo-dynamically produced ROS may cleave the DNA sugar backbone causing single-stranded breaks (SSB) [59] or oxidise guanine nucleotides to produce another photolesion, 8-oxoguanine, which can cause mismatched pairing between the DNA bases [48].

The ROS, ^1^O_2_ and HO· produced by UVR are strong oxidising agents that also target amino acids vulnerable to oxidation, including tryptophan [60], tyrosine [61], histidine [62], cystine [63], cysteine [64], methionine [65], arginine [66] and glycine [67]. For a more comprehensive summary of photo-oxidation of amino acids, the reader is directed to the review by Pattisson et al. [42]. Oxidation-associated changes in protein structure may, in turn, affect function [67,68,69]. UVR exposure can also break or form intermolecular bonds in proteins. In particular, di-sulphide bonded cystine can be reduced to cysteine [56]. These amino acid level changes can affect protein function, with high and low UVR doses decreasing and increasing the thermal stability of collagen, respectively [70,71,72]. UVR can also disrupt the function and structure of lipids via lipid peroxidation [73], resulting in compromised cell membranes. Extracellularly, ROS may cause damage to abundant ECM proteins, such as collagen and elastin [74,75], and to UVR-chromophore-rich proteins, such as fibrillin microfibrils and fibronectin [76]. The differential impacts of UVR exposure on the matrisome (the extracellular proteome) are discussed in detail in Section 4 and Section 5 of this review.

### 2.3. Repair and Prevention of UVR Damage

In response to the damage caused by UVR and/or photodynamically produced ROS, cells can initiate repair mechanisms, including nucleotide excision, to remove photolesions in the DNA [77]. Enzymes in the cell can repair reversibly oxidised proteins, such as cystine, which can be reduced back to cysteine by the thioredoxin reductase system [78], or may break down irreversibly oxidised proteins, typically products of hydroxylation and carbonylation processes [79,80]. In addition, ROS scavengers, such as superoxide dismutases, help restore the ROS balance in the intracellular and extracellular spaces by converting the superoxide anion to hydrogen peroxide [53,81], which is then converted to water and oxygen by catalase and glutathione peroxidase 3 to prevent the formation of hydroxyl radicals [54,82]. We have recently proposed that the biological location of some UVR-chromophore-rich proteins (including β and γ lens crystallins, late cornified envelope proteins in the stratum corneum and elastic fibre-associated proteins in the papillary dermis) may mean that these components act as sacrificial, and hence protective, endogenous antioxidants [76].

## 3. Ionising Radiation (X-rays/Gamma Rays)

In the EM spectrum, ionising radiation is comprised of X-rays (0.01 nm < λ (wavelength) < 10 nm) and gamma rays (λ < 0.001 nm) (Figure 1). Naturally occurring radon gases and cosmic radiation provide a background of ionising radiation of, on average, 2.4 mSv a year [83]. On the other hand, man-made sources of ionising radiation, such as mammography, would commonly only expose the patient to a dose of 0.36 mSv per screening [84,85]. Another key source of man-made ionising radiation that is of particular interest is radiotherapy.

The efficacy of radiotherapy lies in the ability of ionising radiation to penetrate biological tissues, allowing non-invasive targeting and killing of aberrant cells by causing irreparable DNA damage. Historically, radiotherapy utilised naturally occurring sources such as Co-60, which emits 1.2 MeV gamma rays. Modern external beam radiotherapy treatment regimens use linear accelerators (linacs) to accelerate electrons towards a metal target to produce ionising radiation [86], with exposures up to doses of 50 Gy for breast cancer radiotherapy patients [87]. Other forms of radiotherapy include Brachytherapy, where a radioactive source is placed within the patient near the tumour (commonly prostate cancer) site [88]. Inadvertent exposure of healthy tissues along the irradiation path can lead to detrimental side effects, including radiation fibrosis [7] and secondary cancers [89]. While there are newer radiotherapy machines utilising proton or heavy ion beams to reduce exposures to healthy tissue by exploiting the Bragg peak [90] see Appendix A, these treatment options are less widely available and are often reserved for paediatric patients [91]. X-ray/gamma ray radiotherapy remains the foremost therapeutic option, and hence, the impact of these radiation exposures on healthy tissues is a key biological and medical issue.

### 3.1. Absorption of Ionising Radiation (X-rays/Gamma Rays)

In contrast to UVR, photons of ionising radiation are energetic enough to ionise most molecules and atoms [92], potentially leading to the disruption of intermolecular bonds [93]. An abundance of water molecules in biological systems results in a large percentage of ionising radiation being absorbed by water in a process called water radiolysis [94], producing multiple ROS species. Water radiolysis induces the formation of not only hydrogen peroxide, superoxide anion and the hydroxyl radical [57] but also an abundance of highly reactive hydroxyl radicals [95] (Figure 2).

### 3.2. Biological Consequences of Exposure to Ionising Radiation

The exposure of DNA to ionising radiation may directly induce oxidation via deprotonation or electron removal, again producing photolesions such as 8-oxoguanine [96]. Hydroxyl radicals produced from water radiolysis can also disrupt the bonds in the sugar backbone of DNA, resulting in SSBs [49,97]. As ionising radiation is highly energetic, electrons ejected from radical formation could potentially cause further radiolysis of nearby water molecules, resulting in a high density of hydroxyl radicals [95,98], increasing the probability of SSB occurring close enough to each other (within 10 base pairs) to promote the formation of double-stranded breaks (DSBs) [28,99]. DSBs are potentially highly cytotoxic due to the risk of failed repair, such as in non-homologous end joining (NHEJ) or homologous recombination, resulting in gene mutations [100,101], clastogenic effects [102], teratogenesis [103] and carcinogenesis [99].

Ionising radiation-induced water radiolysis can cause significant ROS-mediated damage to proteins through the disruption of peptide bonds, thereby altering their structure and function [67,104,105]. This leads to similar outcomes to those already described in Section 2 including both protein oxidation [106] and lipid peroxidation [107]. The direct impact of ionising radiation on proteins can be observed during X-ray diffraction studies of protein crystals, where cryogenic temperatures reduce the effects of radicals produced by the solvent [108]. These studies demonstrate that di-sulphide bonds and carboxyl groups are most susceptible to localised radiation damage [109,110]. However, this damage may not be evenly distributed throughout the protein [111]. For example, Weik et al. (2000) have shown that the specific disulphide bond between Cys-254 and Cys-265 residues for *Torpedo californica* acetylcholinesterase, as well as the disulphide bond between Cys-6 and Cys-127 for hen egg white lysozyme, are most susceptible to radiation damage. Radiation damage may also localise at active sites in proteins [110,112,113] such as for bacteriorhodopsin [114], DNA photolyase [115], malate dehydrogenases [116], and carbonic anhydrase [117]. This damage localisation has been hypothesised to be mediated either by the presence of metal ions, which have high proton numbers and hence more electrons for photo-absorption to propagate subsequent ionisation events [118], or by the relative accessibility of exposed active sites to ROS [110]. Key extracellular protein targets of ionising radiation are discussed in Section 4 and Section 5.

### 3.3. Repair and Prevention of Ionising Radiation Damage

As both ionising and non-ionising radiation produce ROS, the prevention and repair of damage are largely mediated by the same mechanisms (see Section 2). However, to repair DNA damage specific to ionising radiation, cells utilise base excision repair (BER) for oxidised nucleotides, such as 8-oxoguanine [119,120], while NHEJ and homologous recombination repair (HRR) is activated to remove DSBs [121,122,123].

## 4. Model Tissue Systems for Radiation Studies

Whilst we have discussed the generic cellular responses and molecular damage that both UVR and ionising radiation can cause, different biological tissues may have their own specific responses to radiation exposure. In this review, we have chosen to focus on skin and breast as these organs: (i) are composed of similar tissue types (epithelial, stromal and adipose) (Figure 3) and (ii) have been extensively studied, leading to a comprehensive literature on the molecular and clinical effects of radiation exposure (mainly non-ionising UVR in skin [124] and diagnostic and therapeutic ionising X-rays in breast [125,126]).

### 4.1. Cellular and Acellular Responses of Skin to UVR

Skin is formed of three tissues: the epidermis, dermis, and hypodermis (subcutaneous fat) (Figure 3). The highly cellular epidermis, which is composed primarily of keratinocytes, functions as a key barrier. In contrast, the dermis is vascularised and composed primarily of a complex, collagen and elastin-rich extracellular matrix. Chronic UVR exposure can lead to oxidative stress and increased extracellular ROS in the skin. Cells respond to these stressors with ECM remodelling through the expression of protease, structural ECM components and increased production of protective melanin.

ECM remodelling in photo-exposed skin is complex. However, it is well-established that UVR exposure stimulates keratinocytes to produce inflammatory cytokines [128,129], such as IL-1α and IL-6, and that stimulated fibroblasts express collagen-degrading MMP-1 [130] and MMP-12. This latter protease can degrade elastin [131], resulting in fragmented elastic fibres. UVR exposure also alters the expression of ECM proteins, upregulating the expression of tropoelastin, the soluble elastin precursor [132], but downregulating collagen expression via UVR-induced ROS or mechanical stimuli from the ECM, both of which affect the TGF-β pathway, a well-known control mechanism of collagen production [133,134] in fibroblasts. In the case of ROS, upregulation of CCN1 (cysteine-rich protein 61) [135], which, in turn, inhibits TGF-β signalling by scavenging of TGF-β [136], results in reduced collagen production. A mechanically weakened ECM (as a consequence of protease activity) may also reduce the synthesis of collagen by fibroblasts [137,138], possibly through the downregulation of TGF-β type II receptor [139] or through the upregulation of CCN1 [140]. UVR exposure also reduces the expression of lysyl oxidase (LOX) or lysyl oxidase-like enzymes (LOXL) [141], which are crucial in the facilitation of cross-linking between newly formed elastic fibres [142], contributing to solar elastosis [30]. Further damage may be mitigated by UVR-induced melanin production via the activation of the p53 pathway in keratinocytes [143], which, in turn, stimulates melanocytes in the stratum basale of the epidermis to express the α-melanocyte-stimulating-hormone (α-MSH). α-MSH then upregulates melanin production in melanocytes [144,145].

UVR can also damage the ECM via acellular pathways. Whilst degradation and breakdown of the triple helical structure of collagen can be mediated by photodynamically produced ROS [93,105], high UVR doses and/or non-environmentally attainable wavelengths are commonly required to induce measurable structural and functional effects on fibrillar collagens [124]. We have previously shown [74,76,146] that ECM proteins that are particularly enriched in UV chromophore amino acid residues are susceptible to UVR-induced degradation compared with UV chromophore-poor proteins such as collagen I and elastin. UVR may also affect cross-links between proteins (such as the desmosine-isodesmosine cross-links), which stabilises elastic fibres [147].

### 4.2. Outcomes of UVR Exposure on Skin

UVR penetrating the ECM-rich dermis is linked to collagen degradation, reduced collagen synthesis and disorganisation of elastic fibres. Collectively, this ECM-remodelling has a detrimental impact on the mechanical strength and elasticity of the skin. Specifically, collagen degradation decreases the mechanical tension and stiffness of the ECM, which is hypothesised to reduce the size of fibroblasts, which are less able to exert a traction force on the ECM [93,139]. However, small doses of UVR have also been shown to increase the thermal stability of collagen [71], possibly attributed to the cross-linking of collagen fibres. Elastic fibres in photoaged dermis undergo solar elastosis, which may be mediated by both the degradation and disorganisation of existing elastic fibres and the generation of new, unorganised elastic fibres. Chronic UVR exposure is also associated with an increased risk of skin cancer via, for example, mutations in the p53 tumour suppressor gene [3,143,148].

Whilst clinical outcomes of chronic UVR irradiation on skin are well established, and it is also clear that some abundant dermal ECM proteins are susceptible to remodelling, the functional and structural consequences for the complex skin proteome remain poorly defined.

### 4.3. Structure of Breast Tissue

The composition of breast tissue parallels the composition of skin, but the component tissues are not organised into discrete layers. In the breast luminal epithelial cells line the ducts which are surrounded by myoepithelial cells and encased with a basement membrane made of fibrous proteins, such as collagen IV and laminin, which provide a mechanical barrier [149]. The supporting stroma is comprised of a fibrous ECM made predominantly of collagen I with a cellular population including fibroblasts and myofibroblasts. Finally adipose tissue (which is radio-translucent) is composed primarily of adipocytes [150] (Figure 3).

### 4.4. Cellular and Acellular Responses of Breast Tissue to Ionising Radiation

Human mammary fibroblasts exposed to physiological doses of ionising radiation adopt a senescent-associated secretory phenotype (SASP), enhancing the secretion of ECM-degrading proteases promoting epithelial cell invasiveness and growth in 3D culture [14,151,152]. Key secreted proteases include MMP2 and MT1-MMP1, which drive not only ECM degradation but also cell migration in the basement membrane by exposing a cryptic site in laminin 5 for cell receptors to bind to [153,154]. Ionising radiation has also been shown to activate latent TGF-β1 in the ECM [155], which binds to fibroblasts, triggering their differentiation into myofibroblasts [156].

Ionising radiation can also mediate the release of the growth factors due to ROS-mediated proteolytic cleavage of ECM components [157]. The basement membrane, a key ECM structure that provides structural support to the mammary gland, can further act as a source of matrikines and growth factors, such as the insulin-like growth factor (IGF) [158], which are often sequestered in the ECM. Paquette et al. have shown that reconstituted basement membrane containing these growth factors, when irradiated with ionising radiation, enhanced the invasiveness of breast cancer cells (MDA-MD-231) [157]. The release of other growth factors such as TGF-β1, which is commonly localised in the ECM [159], can also stimulate upregulation of MMPs (e.g., MMP-2, MT1-MMP) in fibroblasts or cancer cells to remodel the ECM [133].

### 4.5. Clinical Outcomes of Ionising Radiation Exposure on Breast Tissue

The biological effects of ionising radiation can be crudely split into two categories: deterministic and stochastic [160]. Deterministic effects are often apparent only when tissues receive high doses of ionising radiation beyond a threshold level [161]. For breast skin, exposure to a dose of more than 6 Gy can induce radiation dermatitis with a severity which is dose-dependent [162]. Such deterministic effects are associated with radiotherapy, where a high dose of ionising radiation is required at the tumour site to trigger apoptosis [163] or necrosis [164]. Consequently, breast cancer radiotherapy patients may experience acute side effects, such as radiation erythema [160] and radiation fatigue [165]. For some patients, late side effects may appear after several months, such as radiation fibrosis, in irradiated regions [7]. Remarkably, there is a lack of evidence for the association between acute and late side effects of radiotherapy in breast cancer [166], and there is little literature investigating the mechanistic understanding that underscores the distinction between acute and late side effects.

The stochastic effects of ionising radiation are probabilistic with a lack of threshold dose [167]. The side effects of ionising radiation from diagnostic breast mammography are therefore skewed towards stochastic effects due to the low doses involved. At low doses, SSB formation is likely the primary trigger for cell repair mechanisms, especially when any DSBs produced are still within endogenous levels. Due to the high fidelity of repair for SSBs [122,168], the occurrence of side effects, which are usually genetic mutations [169], is low. Still, DSBs induced by low doses of ionising radiation can be detrimental. A single unrepaired DSB in a vital gene, such as p53, is sufficient to catalyse tumour growth and mutagenesis [99,170,171], possibly leading to secondary cancers [27]. Similar to skin, the clinical outcomes of exposure of breast tissue to ionising radiation have been well documented, but the underlying mechanisms that elicit such responses have yet to be fully understood.

## 5. The Extracellular Matrix as a Target of Radiation Damage

Whilst the impact of radiation exposure on cells and cellular components is well characterised, ECM–radiation interactions and the downstream biological consequences are not well understood. Crucially, damage to ECM may mediate long-term radiation effects as a consequence of the long half-life and limited repair of key ECM components [172]. The synthesis of many ECM proteins is usually highest during development and diminishes over time [173,174]. Elastin, for example, may persist over the human lifetime [175], whilst dermal and cartilage collagens have half-lives of 15 and 95 years, respectively, [176,177,178]. The slow replacement of damaged ECM proteins would allow changes in the mechanistic signals from ECM to persist, which can lead to long-term complications. Although we have chosen, in this review, skin and breast as model systems to highlight the importance of the extracellular environment and matrix in contributing to the side effects of radiation exposure, there are other organs with important clinical consequences from radiation exposure, of which ECM may also play a role. For example, in breast radiotherapy, excessive ECM accretion may occur in the lung leading to fibrosis [179]. In prostate radiotherapy, ECM degradation often precedes radiation proctitis [180]. For lung radiotherapy, pneumonitis often develops with aberrant ECM deposition [181]. In glioblastoma radiotherapy, the ECM is found to increase the invasiveness of glioblastoma cells, possibly contributing towards the high relapse of glioma patients after radiation therapy [182]. This emphasises the significance and necessity for greater exploration of the ECM following radiation exposure.

To improve our understanding of the repercussions of ionising and non-ionising radiation damage to the ECM, we suggest that future investigations should encompass the impact of radiation on three key mechanisms through which ECM influences cells, namely molecular structural changes, mechanical changes and biochemical changes.

### 5.1. Radiation and ECM Mechanical Properties

UVR and ionising radiation are capable of inducing molecular changes in large ECM proteins, altering their tertiary and quaternary structures, which are essential in maintaining the mechanical properties of the ECM [72,183]. In addition, structural damage to cell-adhesive proteins such as fibronectin could also diminish cell–ECM interactions [184]. This implies that radiation exposure may compromise mechanosensing pathways. Altered tissue stiffness and elasticity may trigger different cellular responses including: initiating epithelial to mesenchymal transition in cancer cells [185,186], triggering senescence in fibroblasts [139,187], determining the fate of differentiating mesenchymal stem cells [188] and even enhancing replication of glioma cells [189]. Determining the mechanical effects of radiation exposure on complex extracellular matrices may provide a better picture of biological radiation response by helping to differentiate between the direct and indirect responses of cells to radiation.

### 5.2. Radiation and ECM Biochemistry

Radiation can also alter the biochemistry of the cellular environment by triggering the release of growth factors that are sequestered in the ECM. Paquette et al. [157] had shown that ionising radiation exposure (20 Gy, Co-60) of Matrigels, which are made from reconstituted basement membranes, released pro-invasive growth factors that enhanced invasion of MDA-MB-231 cells. A plausible mechanism for the release of these factors could be attributed to radiation-induced structural changes to key ECM proteins, such as fibronectin [76,146,190], which binds to a variety of sequestered growth factors, including insulin-like growth factors (IGFs), fibroblast growth factors (FGFs), TGF-β1 and vascular endothelial growth factors (VEGFs) [158,191]. These factors serve as important signals to alter cell behaviour typically via integrins binding [192], MMP-mediated ECM degradation [193] or in wound healing [194]. Radiation damage to fibronectin and other similar ECM proteins may diminish their ability to bind to growth factors, thus increasing the availability of these factors [158] in the extracellular space. Abnormal levels of such growth factors would be taken up by cells, potentially triggering unwanted proliferation and migration due to FGFs [195] or ECM deposition due to TGF-β1 [196].

In addition to growth factors, radiation is also hypothesised to be able to introduce biologically active peptides in the extracellular environment through the fragmentation of ECM proteins. These peptides, often referred to as ‘matrikines’, may be derived from abundant ECM proteins, such as collagen I and IV [197] or elastin [198], and are able to influence cellular behaviour just like growth factors. Whilst there is experimental evidence for the generation of matrikines by MMPs [199] there is a lack of evidence for the direct induction of matrikines by radiation. However, the ability of both non-ionising and ionising radiation to produce ROS that can fragment ECM proteins makes the possibility of radiation-produced matrikines (albeit with less specificity than MMPs), an interesting phenomenon to explore. In all, undertaking these biochemical studies may help explain certain non-local radiation effects, such as bystander effects, where local mechanical influences are not applicable.

### 5.3. Challenges of Studying the ECM and the Current State of Knowledge

Studying the ECM is critical for furthering our understanding of radiation damage, but ECM proteins can be challenging to characterise due to their insolubility necessitating the use of strong dissociative reagents, which may affect protein structure. Secondly, studying the ECM from tissues often requires decellularisation to prevent cellular influence, during which the ECM may be damaged and altered by chemicals used to remove the cells. Various models and experimental systems have been used in recent papers to study the ECM under UV and ionising radiation, but due to their limitations, these systems produce results which can be hard to interpret in relation to other literature. We summarise below the current literature into four general categories of approach, namely: (1) purified proteins; (2) decellularised cultures; (3) ex vivo; and (4) in vivo.

Purified protein experiments (Table 2) represent a bottom-up approach, exploring the effects of radiation on specific ECM proteins that are the building blocks for a complex ECM. Collagen mimetic peptides commonly contain multiple repeats of the tri-peptide sequence (Gly-X-X’) which is a key motif in collagen fibrils [93]. Experiments using these peptides show that molecules rich in the repeating collagen motif are relatively resistant to environmental doses of UVR and require much higher doses (9000 J/cm^2^) [147] or shorter wavelengths (254 nm) [70,200] to elicit changes in their ultrastructure. Higher order proteins structures are however important as purified collagen gels, exhibited increased stiffness and reduced elasticity after exposure to physiological doses of UVB [201]. Other ECM proteins, such as fibronectin and fibrillin, are also found to be structurally susceptible at physiological doses. In contrast with UVR exposure, X-ray studies on purified collagen show more prominent structural changes and some specificity in peptide bond cleavages, but these studies often utilise non-physiologically relevant doses in the range of 100k Gy. Interestingly, the study by Miller et al. (2018) [183] showing that both acute and fractionated doses of ionising radiation on collagen gels did not differ in their results, indicating that dose rates might not affect the outcome of radiation exposure, at least in purified protein systems. However, dose rates can significantly affect acute/immediate physiological responses in more complex systems involving cells (in vivo or in vitro) [202], such as FLASH radiotherapy [203], implying that dose rates could be a key variable when looking at how a complex tissue may remodel the ECM. The advantages of using purified proteins are to isolate radiation effects on peptides, allowing a greater look at the chemical and structural changes that might occur during irradiation through mass-spectrometry or X-ray scattering. However, the results are difficult to interpret with regards to tissues and the potential downstream effects in vivo as the experimental system is not representative of the protein’s natural environment or state in the ECM. Nonetheless, such reductionist experiments provide a fundamental picture of the molecular mechanisms that occur for the proteins in question during radiation exposure and possibly allow us to piece together evidence from individual ECM proteins to predict or explain complex phenomena.

Decellularised cultures (Table 3) involve taking tissue samples from living organisms and removing the cells from the tissue, leaving behind the ECM scaffold and proteins for cell culture applications [206,207]. There is still little evidence of UVR affecting large scale mechanical properties of such ECM scaffolds, although localised changes in molecular structure may be occurring [208]. There is, however, evidence of X-rays, again in the 100k Gy range, reducing or increasing stiffness of these ECM scaffolds. Behaviour of cells seeded onto X-ray irradiated ECM scaffold was also altered with increased proliferation [206] or poor adhesion [209]. Overall, these studies show that X-rays have a higher propensity than UVR to induce changes in mechanical properties of ECM scaffolds and that cellular behaviour can be affected. 

Utilising ECM from tissue provides the advantage of good in vivo representation, with ECM structures, growth factors and binding ligands largely intact for radiation studies. Decellularised cultures allow us to look at the interplay between different components of the ECM, as well as post-irradiation cellular remodelling of the ECM. However, the abundance of ECM components in the culture can also cause problems when attempting to identify the cause of downstream effects. The variability of ECM proteins in different organisms, or even in different regions of the same organism, can also make the experiments difficult to replicate. Lastly, the decellularisation process can also alter ECM protein’s ultrastructure during the chemical removal of cells or during the sterilisation process [210,211].

Ex vivo systems (Table 4) refer to tissues that are extracted from organisms and are cultured with minimum alteration. Experiments with ex vivo samples are typically conducted less than 24 h after biopsy to prevent influence from external sources. Such experiments have mostly shown a reduction in mechanical strength, even at radiotherapeutic doses of X-rays.

A key advantage of ex vivo experiments is their ability to provide insights into complex tissues, making them useful for determining end point consequences of radiation effects. However, these systems are often complicated to analyse as they contain both ECM and cells that can influence the remodelling of the ECM after irradiation. Further, the results are hard to generalise as the tissues used are made of specific cells and ECM environments, which may only be applicable to that organism.

For in vivo systems (Table 5), mouse models are often used to observe longer-term tissue responses to radiation. In addition, such models can elucidate if irradiated regions may exert a local or systemic influence on, for example, the immune responses [215]. In vivo studies show that irradiated animals experience ECM remodelling, which is likely to be mediated by MMPs. Such experiments are useful to account for the various effectors of radiation response by allowing full interaction between different mediators and are helpful for observing long-term effects such as secondary malignancies, fibrosis and metastasis. However, therein lies the challenge of relating animal models back to humans, as genetic differences could invalidate the radiation response elucidated in these models [215]. In addition, it is difficult to determine if the radiation outcomes are associated with the acute effects of radiation on ECM proteins or with long-term remodelling from cellular expression of MMPs.

The various types of experiments summarised above show evidence that ECM components are differentially affected. At the molecular scale, purified ECM proteins show alteration of individual protein functions and structure, while ECM mechanical properties are altered at a larger scale in decellularised cultures and ex vivo. In addition, in vivo experiments demonstrate remodelling of ECM and abnormal cellular behaviour following irradiation. Still, although effects of ionising and non-ionising radiation are observed on different scales, they are not yet integrated into a coherent understanding of the underlying mechanism of radiation damage. Furthermore, most studies using purified ECM proteins and decellularised cultures focused on high doses that are valid for sterilisation processes but are not clinically relevant.

To address this gap, it is necessary to have a consistent methodology that allows investigation on all scales (from molecular to in vivo tissues) to provide a link between localised molecular changes in ECM proteins and the transformation of global properties such as mechanical strength. One approach that our group is undertaking is peptide location fingerprinting [74], which takes advantage of the sensitivity of modern mass spectrometers, allowing us to probe the ECM proteins of interest both in purified samples and in complex tissue. We have previously shown that this fingerprinting approach was able to distinguish between fibrillin derived from the eye and from skin despite having similar peptide compositions [222] and that physiologically relevant doses of UVR can induce statistically significant differences in the yield of ECM peptides (as detected by mass spectrometry) in multiple proteins [74]. We have recently shown that the same technique can detect multiple putative biomarkers of skin photoageing [223]. Structural changes identified by this technique could be key to studying the effects of ionising radiation on ECM proteins and scaffolds at physiological doses where the outputs are often less apparent. Another approach could be in vitro models that better mimic the in vivo environment. Culturing cells in 3D, for example, has been shown to more closely replicate not only tissue composition [224] but also structure [225]. Utilising such approaches may help to model not only radiation-induced damage but also in the development of strategies to protect and repair tissues.

## 6. Conclusions

Many studies in the past exposing the detrimental effects of UVR and ionising radiation were focused on modifications of cellular behaviour as a consequence of intracellular molecular damage to DNA, proteins and lipids [99,226,227]. Additional study of ECM proteins would be beneficial, increasing our understanding of sub-lethal radiation responses that are often long-term and detrimental to the quality of life. Both skin and breast tissue provide clinically relevant models to study the biological effects of UVR and ionising radiation, respectively. There remain gaps within the current literature on the understanding of the molecular mechanisms that trigger these unwanted radiation side effects. Damage to the ECM, an important yet overlooked mechanical and biochemical regulator of cellular processes, could be key to understanding these clinical consequences. The ECM is vulnerable to accumulating radiation damage due to a lack of a robust repair mechanism, compounded with a long turnover rate for most ECM molecules [175,178]. This may trigger mechanosensitive pathways [137,139,228] or release innate growth factors that were bound to the ECM, triggering downstream effects. Current published studies (which involve studying purified ECM proteins, decellularised cultures, ex vivo and in vivo systems) lack a coherent link to capture the underlying mechanism from radiation damage to clinical outcomes. Future studies to address such a gap should aim to exploit systematic approaches that can be applied at the different scales of ECM experimentation, whilst using comparable physiological doses of radiation, to draw possible connections between radiation damage and clinical outcomes of radiation exposure. This has the potential to enhance our insight into the origin of radiation damage and thereby allow us to make better predictions for outcomes to radiation exposure.

## Figures and Tables

**Figure 1 cells-10-03041-f001:**
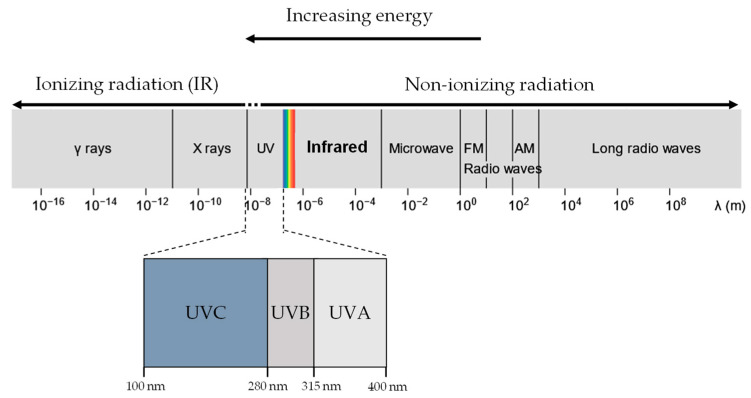
UVR, X-rays and gamma rays all lie in the electromagnetic spectrum. UVR (UV-A and UV-B) lie at a slightly higher energy range compared to visible light and are generally considered non-ionising. In contrast, X-rays and gamma rays have much higher energy than UVR and are considered ionising radiation.

**Figure 2 cells-10-03041-f002:**
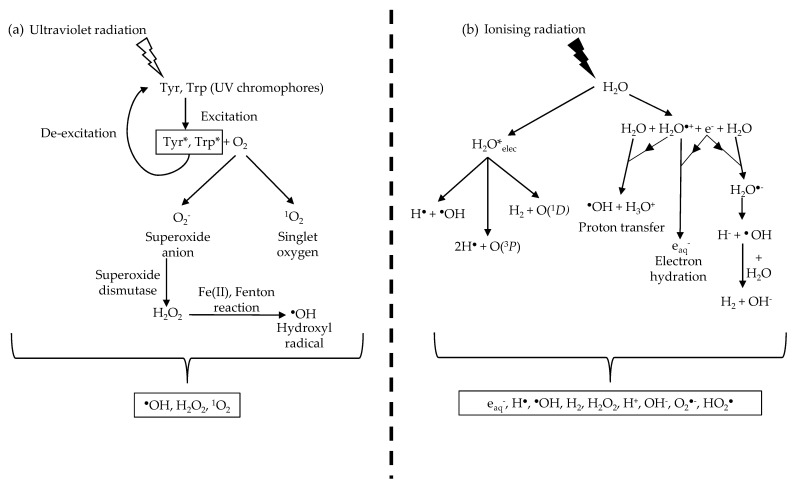
UVR and ionising radiation indirectly damage biological molecules by ROS production. (**a**) UVR produces ROS through UV chromophores that absorb UVR and undergo excitation. The excited chromophores react with oxygen molecules to form singlet oxygen and the superoxide anion. The superoxide anion is converted to hydrogen peroxide by superoxide dismutase before undergoing the Fenton reaction in the presence of Fe (II) to form the hydroxyl radical. (**b**) Ionising radiation produces a range of ROS and, more crucially, the hydroxyl radical through water radiolysis. This results in a larger concentration of hydroxyl radicals produced during ionising radiation irradiation compared to UVR due to the abundance of water molecules. Information from Figure (**b**) was sourced from Meesungnoen J. et al. [57].

**Figure 3 cells-10-03041-f003:**
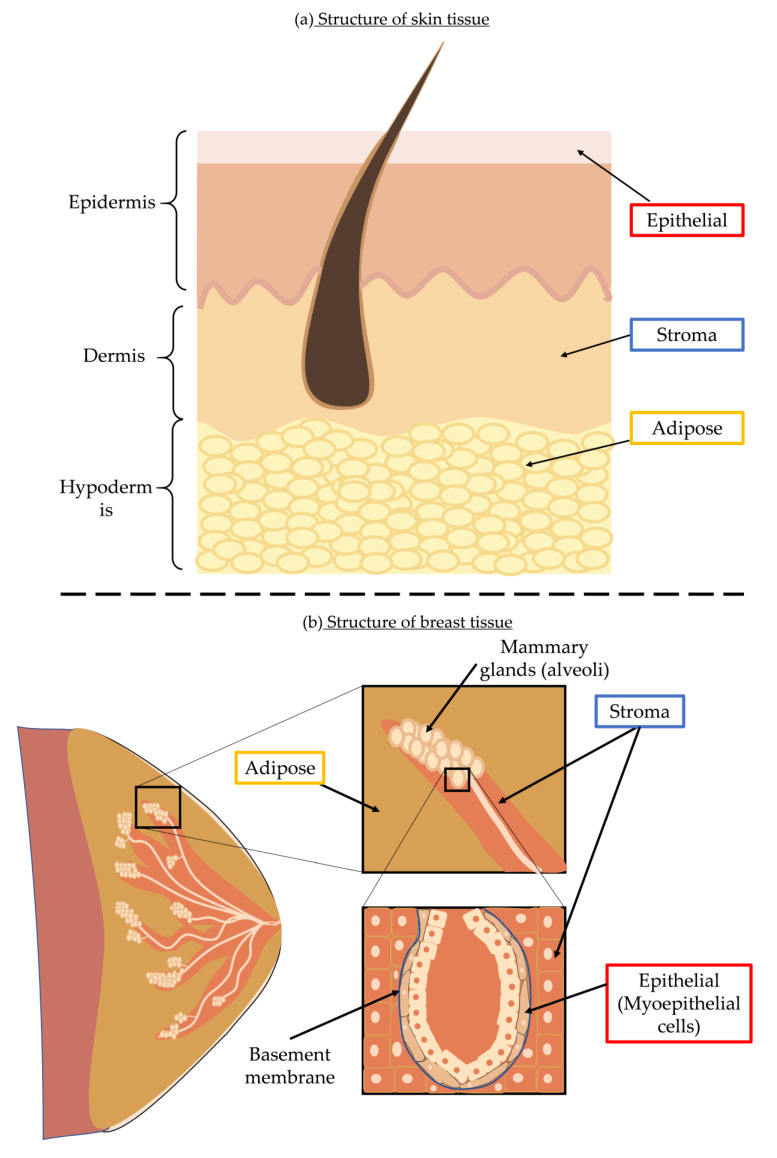
Tissue composition of skin and breast. (**a**) The skin is composed of an outer epidermis (containing epithelial cells), the dermis (containing stromal cells) and the subcutaneous layer (hypodermis) containing adipocytes. (**b**) Breast tissue contains epithelial cells that make up the alveoli structures, adipose tissue, as well as connective tissues or stroma. This structural composition is akin to skin, albeit with different spatial distribution but serves comparable purposes and could be similarly affected by radiation [127].

**Table 1 cells-10-03041-t001:** Human exposure to ionising and non-ionising electromagnetic radiation can come from the environment or from clinical interventions. Exposure to both types of radiation can have clear clinical benefits but may also result in detrimental biological effects.

	Type	Environmental Exposure	Clinical Exposure	Biological Consequence
**Ionising**	X-rays/Gamma rays	Cosmic radiation [10], Radon gas [11]	Diagnostic imaging [12], Radiotherapy [13]	Fibrosis [14], Carcinogenesis [15]
**Non-ionising**	UVR	Sunlight [16]	UVR Phototherapy [17]	Skin photoageing [18], Vitamin D synthesis [19]
	Visible light	Sunlight [20]	Photodynamic therapy [21]	Ocular phototoxicity [20]
	Infrared	Sunlight [22]	Neural stimulation [22]	Skin photoageing [23]
	Radiowaves	Lightning [24]	Hyperthermia [25]	Brain activity [26]

**Table 2 cells-10-03041-t002:** Selected studies utilising purified ECM proteins for radiation damage experiments are useful to elucidate molecular mechanisms of radiation responses for individual ECM proteins. Most studies show that purified collagens in solution are relatively resistant to UVR at physiological doses which can affect other ECM proteins. Ionising radiation, however, induces significant structural change and peptide bond cleavages albeit at at non-physiological doses.

Radiation	Dose	Method	Ref.	Results
UV	UV (254 nm), 24.0 J/cm^2^, 102.0 J/cm^2^, 396.0 J/cm^2^	Collagen model peptides and rat tail tendon collagen I	[200]	Rat tail collagen exhibited stable intermediate after irradiation. Gly-Pro-Hyp mimetic collagen was more stable than Gly-Pro-Pro, while Gly-Ala-Hyp was more stable than Gly-Pro-Hyp.
	UV (254 nm), 5–187 J/cm^2^	Sterile rat tail collagen I	[70]	Collagen denatures with loss of hydrogen bonds with water molecules, followed by the loss of triple helix and peptide bond cleavage.
	Broadband UVB (270–380 nm) 0.1 J/cm^2^, Solar radiation (SSR), 30 J/cm^2^	Fibrillin/Collagen VI microfibrils derived and purified from human dermal fibroblasts. Peptide mass fingerprinting	[74]	No changes for collagen IV. UVB/SSR increased protease susceptibility for fibrillin, possibly from ultrastructural changes.
	Broadband UVB (290–320 nm) 3.2–9.6 J/cm^2^, BL/DMR lamps (320–400 nm) 49–147 J/cm^2^	Bovine dermis native collagen, made into collagen gels using sodium bicarbonate	[201]	UVR at 300–340 nm caused hardening and reduced elasticity of collagen gels, and 330 nm gave the greatest effect. Increase in tyrosine cross-linking was found.
	UVB (280–315 nm), 20–500 mJ/cm^2^	Purified collagen-1, fibrillin microfibrils from biopsy/COS-1 cells, fibronectin from bovine plasma	[146]	UVB dose required to damage ultrastructure decreases with greater chromophore composition. Collagen I was the most UVB-resistant, followed by fibronectin and then fibrillin.
	UVA (365 nm). 9330 J/cm^2^	Isolated bovine nuchal ligament elastic fibres	[147]	No ^13^C NMR shifts detected, 11% reduction of desmosine from cross-link cleavage.
Ionising radiation	Co-60 γ-ray at 1.289k Gy/h, 5k–50k Gy	Lyophilised collagen from rat tail tendon irradiated and tested for solubility and melting temperature	[204]	Irradiated samples were, in general, more than twice as soluble as non-irradiated in 0.02 M acetic acid, 6 M lithium chloride and 6 M urea. Melting temperature reduces with increasing dose.
	γ-ray (1 MeV), 50k–500k Gy	Grounded collagen irradiated in dry/wet (5%/80% moisture) state in the presence and absence of oxygen/nitrogen	[205]	Solubility unchanged when irradiated wet due to cross-linking, and solubility increased when irradiated dry. Significant molecular changes likely due to the breakage of peptide bonds. Degradation of Tyr; Hyp/Pro; Asp sensitive to oxygen/nitrogen.
	Near X-ray (13.8–22.1 eV)	Isolated collagen mimetic peptides, photon absorption in gas phase + mass spectrometry	[105]	Gly-Pro peptide bonds are more susceptible to cleavage, collagen triple helix stabilised by hydroxyproline.

**Table 3 cells-10-03041-t003:** Decellularised tissues exemplify a highly representative ECMs that can mimic key asepcts of in vivo responses to radiation damage. Studies using these systems show that X-rays have a higher propensity than UVR to induce changes in mechanical properties of ECM scaffolds, and that X-ray exposure can affect subsequent cell responses to the ECM.

Radiation	Dose	Method	Ref.	Results
UV	UV (254 nm) using UV cross-linker, 2 cycles (90 s each)	Decellularised Lewis rat intestines	[208]	No significant change in collagen/GAG content. Loss of villous ECM projections.
	γ-ray (wavelength unspecified, 5000 Gy)	Rabbit kidney decellularised	[210]	Reduced tensile strength and young’s modulus with gamma ray.
Ionising Radiation	Co-60 γ-ray, 25k Gy	Gamma irradiation of decellularised cornea	[211]	Increased stiffness/tensile strength, reduced elongation at break after irradiation, due to fragmented collagen cross-linking.
	Cs-137 γ-ray, 1k–10k Gy	Decellularised whole porcine kidney	[209]	3k Gy resulted in more than 50% loss in collagen content. Human renal cortical tubular epithelium (RCTE) cells reseeded and resulted in poor adhesion/growth.
	Cs-137 γ-ray, 20 Gy	Murine mammary fat pads decellularised and made into hydrogels.	[206]	Increased proliferation for murine TNBC reseeded on irradiated hydrogel.

**Table 4 cells-10-03041-t004:** Ex vivo experiments utilise complex model systems that give biologically relevant consequences of radiation effects. Studies show that radiotherapeutic doses of X-rays (around 50 Gy) can alter the mechanical properties of ex-vivo samples.

Radiation	Dose	Method	Ref.	Results
UV	UVA (365 nm, 1.5 mw/cm^2^), UV-B (302 nm, 1.6 mw/cm^2^), UV-C (265 nm, 1.8 mw/cm^2^), dosage: 10–4000 J/cm^2^	Stratum corneum from breast skin tissue extracted	[18]	Reduced stiffness, fracture stress/strain, at >4000 J/cm^2^ UVA and >400 J/cm^2^ UVB. The energy required to fracture decreases in a dose-dependent manner.
	Cs-137 γ-ray, 10–63 Gy	Mammary tumours (MMTV-PyMT transgenic mice) immediately irradiated and frozen before tested for compression	[183]	Significantly reduced tensile and compression modulus after 60 Gy irradiation (fractionated and single dose).
Ionising Radiation	6–10 MeV X-rays, 30–56 Gy	Biopsy from radiation therapy treated breast cancer patients. Irradiated/non irradiated samples from the same patient 10-96 months after treatment	[212]	No observable change in elastic fibres/collagen, but stiffness is higher for irradiated regions.
	21 KeV X-rays, 50–35,000 Gy	Lumbar vertebrae excised and removed of soft tissue. Wrapped in saline-soaked gauze	[213]	Monotonic strength (one direction) decreased at 17,000 Gy and above. Increase in non-enzymatic cross-links at a lower dose (50–1000 Gy) by analysing AGEs. Crosslinks do not have a significant impact on vertebral strength.
	6 MeV X-rays, 10–100 Gy	Bovine pericardial tissue (collagen), Bovine ligamentum nuchae (elastin)	[214]	For pericardial tissue, elastic modulus increased for small strain and decreased at larger strain after irradiation. Elastin has significantly reduced elastic modulus after irradiation.

**Table 5 cells-10-03041-t005:** In vivo models allow for observations of long-term radiation responses not only in the targeted area but also surrounding tissues or organs for bystander studies. In vivo studies showed that both UVR and ionising radiation-exposed animals experience ECM remodelling as a consequence of protease action.

Radiation	Dose	Method	Ref	Results
UV	UVB (285–350 nm, peak: 310 nm). 0.12 J/cm^2^, (MED) ×3/×6/×9 per week for 13 weeks	Skh1/Hr female mice irradiated with UVR over 13 weeks with increasing dose	[216]	KPA inhibited cathepsin G, which mediates MMP-1 upregulation through Fn fragmentation/activating pro-MMP-1.
	UVA/B (240–320 nm), 1 MED	Albino guinea pigs (400–500 g) irradiated on shaved skin and decapitated 2/4/72/192 h after irradiation	[217]	Disorganisation of collagen I/II fibres worsen over time. Increase in collagen III detected.
	UVB (280–320 nm, peak: 313 nm). 0.08 J/cm^2^ (1 MED), 3 times per week for 20 weeks	Skh1/Hr female mice 8 weeks old irradiated with UVR over 20 weeks and allowed to recover for 10 weeks. Dorsal skin biopsies were taken at week 28 and 38	[218]	After 20 weeks of irradiation, there was a 35% reduction in collagen content. Collagen further declined during recovery by ~70%. mRNA levels of MMP-3 and 9 were not regulated, while mRNA of MMP-13 decreased. Possible degradation of collagen by the activation of latent MMP rather than increased expression.
	Co-60 γ-ray, 2–22 Gy in fractions of 2 Gy/day	White, outbred rats, irradiated in bladder and rectum. For 2 Gy, rats were harvested 1 day/1 week/1 month after irradiation. Higher doses harvested after 1 day.	[219]	One-month post-2 Gy irradiation showed thickening of collagen fibres and tight, parallel packing for the bladder and rectum. One day post-irradiation for higher dose observed the same effects with the severity dependent on dose. Skin most sensitive showing similar damage at 8 Gy.
Ionising Radiation	300 kVp X-rays (30–60 Gy) for local, Cs-137 γ-ray (6–10 Gy) for whole body	C57BL/6 mice with smad3 gene knockout	[220]	Smad3 knockout mice have less TGF-β1 expression, less inflammation, less myofibroblasts after radiation
	Co-60 γ-ray, 2–40 Gy, 1.7 Gy/min	2-month-old, white wild type outbred rats, ~ 200 g, harvested 1 day/1 week/1,2,3 months after irradiation for rat’s tail tendon	[221]	Differential scanning calorimetry showed a dose-dependant increase in denaturing temperature 24 h after irradiation, but dose-independent after 1 week. Negligible change was observed for tertiary/secondary structures using second harmonic generation/cross-polarisation optical coherence tomography

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
