# Peer review of "The Effects of Ionising and Non-Ionising Electromagnetic Radiation on Extracellular Matrix Proteins"

_cells, 2021, doi:10.3390/cells10113041_

Round 1

Reviewer 1 Report

The review by Tuieng et al., summarizes the literature on the biological effects of ionizing and non-ionizing radiation on the ECM of breast stroma and skin dermis. Overall, the text is fairly well written and because it is aimed to focus mainly on extracellular matrix (with less focus on cells), it has novelty.  

However, again because the authors chose to cover both ionizing and non-ionizing radiation the focus gets dispersed in parts of the text.  The limited literature on radiation induced non-cell biological effects also hinders the focus (and in many parts of the text the authors talk about cellular effects of the ionizing and non-ionizing radiation).

Specific Comments:

1- Instead of covering the cellular effects of tradition, perhaps try to be more bold on interpretation/speculation of how alterations in ECM via non-ionizing and ionizing radiation would effect normal tissues/cells in not only skin and breast tissue but other organs (CNS, reproductive, lung etc)

2- Make sure to include more information on terminology (such as Bragg peak) for the audience, who may not be as familiar with these definitions.

3- Although the therapeutic use of protons and heavy ions are less common, they are still relevant not only their use in cancer therapy but also their abundant existence in space environment (Galactic Cosmic Rays).  Regardless how limited the literature on this, that would be great to see their direct effects on ECM and how these effects differ from x-rays and gamma-rays.

4- Are there any studies looking at the how changes in dose rates affect the alterations in ECM following ionizing radiotn exposures?

Reviewer 2 Report

This narrative review examines effects of EMR, both non-ionising and ionising, on ECM proteins.

I’m not sure that Figure 1 is required.

Table 1: Radon, eg, from cement walls, should be added, beneath cosmic radiation exposure.

Table 1: what about including radio waves and radar in Table 1, they are also EMRs.

Line 99: there are other cellular targets of IR than DNA, eg, cell membranes in lymphoid cells.

Line 101: modern treatment regimens also use brachytherapy and unsealed sources.

Line 102/3: re 50Gy. Often, higher doses are used, eg 60 or even 70Gy.

Line 102: the metal target in a Linac head is routinely removed to produce electrons.

Line 141: an abundance.

Line 166: re DSBs: could also mention here that they are clastogenic and teratogenic.

Line 167: mention homologous recombination also here.

Line 279: A stronger case needs to be made as to the relevance of the model systems selected for focus, particularly breast tissue. Attempts have been made to underline the similarities between breast and skin as model systems, but they are clearly more dissimilar than similar. If the reason for selecting breast and skin is simply that that is where the data is, it should be stated. Why not just concentrate on skin as a model system for both UV and IR?

Line 311: 6Gy, especially if fractionated, shouldn’t cause radiation dermatitis.

Line 315: breast cancer patients don’t get acute radiation sickness.

Line 323: re SSBs being “the main DNA damage at low doses..”. This is debatable, it depends how you define “main”. SSBs and DSBs are formed in similar proportion per radiation dose.

Line 316/7: radiation fibrosis may occur after radiotherapy, but there is a weak association between the occurrence of acute severe effects and late effects such as fibrosis. Consequential late effects can occur in some breast cancer patients. This should be discussed.

Line 327/8: sense?

Line 365/6: what is the evidence for IR-induced release of ECM-sequestered growth factors? No citations are provided. This is a key element in the reasoning as to how IR might induce functional cellular effects such as migration and proliferation.

Line 508/9: re “physiological doses” of radiation. This is another key issue in terms of the relevance of any of this research to the clinical situation. If supralethal doses of radiation are required for experiments to provide registrable outputs, then calling for the use of physiological dose research is pointless unless better model systems are available. The authors should further elucidate what such systems might look like and what would need to happen to facilitate their development. Until then, the relevance to clinical endpoints of supralethal IR effects on the ECM will remain obscure.

Round 2

Reviewer 2 Report

Good job on the revision, thank you.